# Perceived discrimination and contextual problems among children and adolescents in northern Chile

**Jerome Flores**[1☺]*, **Alejandra Caqueo-Urízar**[2☺], **Lirna Quintana**[1‡], **Alfonso Urzúa**[3‡], **Matías Irarrázaval**[4‡]

**1** Centro de Justicia Educacional y Escuela de Psicología y Filosofía, Universidad de Tarapacá, Arica, Arica y Parinacota, Chile, **2** Instituto de Alta Investigación, Universidad de Tarapacá, Arica, Arica y Parinacota, Chile, **3** Escuela de Psicología, Universidad Católica del Norte, Antofagasta, Antofagasta, Chile, **4** Departamento de Psiquiatría, Facultad de Medicina, Hospital Clínico Universidad de Chile, Santiago, Región Metropolitana, Chile

☺ These authors contributed equally to this work.
‡ These authors also contributed equally to this work.
* jflores@uta.cl

**Data Availability Statement:** All relevant data are within the manuscript and its Supporting Information files.

## Abstract

Discriminatory behaviors among inter-ethnic relations in schools have long been noted and studied, but there are several correlations between discriminatory behaviors and other constructs that need further investigation. As an example, the relation between perceived discrimination and contextual problems—which include family, school and peer problems—among children and adolescents in Latin America has received little attention from previous studies. Further, the mediating role of ethnic identification and collective self-esteem in this relation also needs to be considered as they could be proven as protective factors for discriminatory behavior and its outcomes. Therefore, this study aimed to, first, establish the relationship between perceived discrimination and contextual problems in inter-ethnic students aged 8–19 years living in Arica, Chile; and second, to identify the role that ethnic identification and collective self-esteem play within this relation. In order to investigate this matter, a cross-sectional study was carried out with 3700 students in 29 schools between the fourth year of primary education and the last year of secondary education, aged between 9–18 years, with 48.4% men and 51.6% women. The sample was divided into primary and secondary school groups. The scales utilized were the Everyday Discrimination Scale, Multi-Group Ethnic Identity Measure-Revised Scale, Collective Self-Esteem Scale and the dimensions of contextual family, school and peer problems, as well as the general index of contextual problems of the Child and Adolescent Assessment System. For data analysis, we tested a path analytic model at both the within and between levels to account for the relations between variables. In each group the models obtained an optimal fit. We found that perceived discrimination and ethnic identification were directly related to contextual problems (.23-.39), and collective self-esteem had only a mediating role. This study showed that strategized interventions focusing on ethnic identification and perceived discrimination should be utilized by schools to create a better developing environment.

**Funding:** This research was funded by ANID PIA CIE160007.

## Introduction

It has been reported that migration to Chile has been increasing since 2015, with 14.8% of it being concentrated in the three northernmost regions of the country [1]. Further, according to data gathered from the Centro de Estudios del Ministerio de Educación de Chile [2], the number of foreign students has increased, and enrolment of foreign students is concentrated in the three northernmost regions, with 35% of the national total in 2015, and 30.7% in 2017.

As a result of this immigrational process, the schools within bordering cities face the constant challenge of including students from various nations into their educational system [3], and this is in consonance with the rights of the child determined by the United Nations, as it states that access to education should be guaranteed to children all around the world [4]. Therefore, in the three northernmost regions of Chile, over 80% of schools have foreign students enrolled [2], and this situation already poses a very complex problem for schools to handle as they have to deal with numerous cultural and ethnic differences within the same schooling environment, but this fact may become even more complex when taken into account that there are also native populations (from border cities) also enrolled in these schools, such as the Aymara in northern Chile [5].

Deepening this discussion, previous studies have shown that both migrant and ethnic minority populations have long been victims of discrimination [6, 7], and a study performed in Chile addressing inter-ethnic relations established that perceived discrimination plays a key role in how these relations develop [8]. This is not a surprise, as the Arica and Parinacota regions of Chile are subjects of many diverse and multicultural contexts, and the development of these relationships is almost inevitable.

Regarding the discriminatory process, it has been noted that the experience of discrimination can affect mental and physical health in multiple ways during childhood and adolescence [9–11], and is a predictor of antisocial behavior and substance use [12]. However, previous studies have paid little attention to discrimination in the infantile-juvenile population in Chile, and some of the existing studies have solely focused on racial discrimination related to the Mapuche population and the Peruvian migrants [13, 14].

Furthermore, when considering the previous statements, it is not difficult to imagine that there may be students suffering daily with discrimination within schools in the northernmost region of Chile, and so it is possible to infer that there are constant tensions being generated in their relations with family, school, and peers since discrimination may affect children's perception of various aspects of their environment. It has been shown that discrimination affects family conflict, which in turn can mediate the relation between discrimination, well-being, and mental health in different populations [15–19]. Further, a negative correlation has been found between discrimination and students' perceived value/utility of their schools [20].

In addition, in Bronfenbrenner's ecological theory, three constructs were created to better conceptualize a child's environmental relations: the child's immediate environment includes his/her family and school, which constitute parts of the child's environmental microsystem; the interactions between these two microsystems comprise parts of the mesosystem, and the child's social values and various cultural aspects constitute parts of the macrosystem. A key aspect of this theory is the notion that the child is an active subject that is constantly interacting with his/her environment through progressive changes and adaptations, and that these changes and adaptations may vary in scope and form [21, 22].

Coinciding with this conceptualization, in a review study that analyzed the influences that the schooling cultural context may have on adolescents' development, it was concluded that most researches consistently demonstrate that adolescents actively construct their identities through social interaction; that these social interactions, in turn, are defined by the

environments they inhabit; that these environments are either chosen by them or they are allowed to participate in; and, ultimately, that these constructed identities are critical to all aspects of their development [23]. In addition, a review of studies on adolescent development in Latin America highlighted that the role of school and peer relations is a fundamental element in understanding children's emotional development and in addressing their ethnicity, and this may happen because there may be experiences of discrimination and exclusion within these relations that have been made invisible to others [24].

From an ecological-transactional perspective, a study demonstrated that attention should be directed to the child's family, school, and peer relationships, so as to better understand how discrimination can affect the child's perceptions over these relations, and to better investigate the contextual problems that permeate them [25]. In terms of conceptualization, contextual problems refer to conflicts in the three main environmental relations that children and adolescents experience during their development: family problems, school problems, and peer problems. Further, it is also necessary to clarify the relation between perceived discrimination and contextual problems as a better understanding of this relation could be turned into strategized interventions to help mitigate discriminatory behavior.

Previous studies have also taken efforts to better conceptualize the relation between discrimination and well-being, and at least two mediators have been proposed: collective self-esteem and ethnic identification [15, 26]. In concept, collective self-esteem refers to an individual's own assessment and perceptions regarding others' assessments of the group to which that individual belongs [27]. Regarding this construct, a previous study found that collective self-esteem has a mediating effect between perceived discrimination and emotional well-being in adolescents [26]. Moreover, the findings of another study showed that perceived discrimination has a negative effect on collective self-esteem [28], a fact that requires further investigation to ensure the validity of the correlation.

In concept, ethnic identity is defined as "that part of an individual's self-concept that derives from his or her knowledge of belonging to a social group, along with the evaluative and emotional meaning associated with that belonging" [29]. Additionally, this belonging can be linked to different social groups, including nationality, ethnicity, religion, among others. According to previous research, this mediator can be a protective factor against the negative consequences of many types of discrimination [27, 30–32]. The concept of ethnic identification is used in this study because ethnic identity is simply a way of labeling ethnicity.

However, it may not have the same effect on different ethnicities since some findings show that it may have a significant effect on African American's mental health, but not on Latin Americans [33]. It has also been observed that the lack of ethnic identification can have a detrimental effect, exacerbating somatic symptoms in the face of high cultural and educational stress [34]. In addition, in Chile, ethnic identification has been negatively correlated with the presence of antisocial behaviors [35], and the results obtained in another study suggest that ethnic identification is key to the maintenance of the Mapuche culture [36]. Additionally, it has been found that ethnic identification can predict collective self-esteem [37].

Based on these inferences and previous knowledge, we believe there is a need for further investigation to establish whether the two variables, ethnic identification, and collective self-esteem, can mediate the relationship between perceived discrimination and contextual problems. This is factual because they have been observed as mediators in other close relations—such as between discrimination and psychological well-being—and because both seem to be directly linked to children' s/adolescents' identities. Thus, they are regarded as potential protective factors for discrimination.

In summary, there is a lack of studies in the relation between perceived discrimination and contextual problems in Latin America, especially in the children and adolescents population,

and the role that collective self-esteem and ethnic identification may play within this relation is also unknown. Based on the aforementioned facts, we believe this study could contribute significantly to a better understanding of the effects of discrimination in schools within the northernmost regions of Chile and other applicable contexts.

Finally, the aim of this study was to, first, establish the relationship between perceived discrimination and contextual problems in inter-ethnic students aged 8 to 19 living in the northern region of Chile; and second, to identify the role that ethnic identification and collective self-esteem play within this relation. Three hypotheses are posed from the objective: 1) The greater the perceived discrimination, the greater the contextual problems. 2) Ethnic identification has a mediating role between perceived discrimination and contextual problems. 3) Collective self-esteem has a mediating role between perceived discrimination and contextual problems. Furthermore, age and sex were controlled in these relationships.

The relationships expected to be found from hypothesis are presented in the theoretical models in Figs 1 and 2. The first considering contextual problems in a global way. The second considering contextual problems divided into three: family, school and peers. Both are tested in primary and secondary. Since vulnerability is a variable that is measured at school level, it is only considered at a second level.

## Methodology

The Ethics Committee of Universidad de Tarapacá approved this research. parental consent was in writing. student consent was in writing.

This is a non-experimental study because variables were not manipulated, and we used a cross-sectional correlational-causal approach, as we measured all variables in a single analysis.

## Participants

The initial sample of this study included 3923 students from 29 schools situated in Arica, Chile, between the fourth year of primary education and the last year of secondary education, and the age range of these students was from 8 to 19 years. The sampling was done for convenience. It is a sample of the general school population, and there is no information on I.Q. or diagnosis of previous mental disorders. SENA has different versions according to the age of the participants. While the contextual problem scales in both share the wording of most of their items (77%), varying the rest to suit age, and the secondary version has 1 item more than the primary version. It was then considered more accurate to divide the sample according to the version being used and to analyzed the two groups separately.

The division of the sample into primary and secondary groups is made on the basis of the grades in which the students are found, with the 8–12 years old version applying to fourth, fifth and sixth grades of primary and the 12–18 years old version applying to seventh until to last year of secondary. Originally the range of primary is from 8 to 11 years, but finally, the eight years old students were excluded as there were only 22 cases. Although 12-year-old students may answer the primary instrument, they were excluded from the sample to avoid confusion with 12-year-old students who answered the secondary version. There were no exclusion criteria other than the approval of the students' parents or guardians to participate, as well as the students' own willingness to participate.

To determine the final sample, were discarded 1.8% of the students because they did not properly complete the SENA and the automated online correction of this instrument does not allow more than 10% of unanswered items. Multiple imputation (IM) was used for the missing values since is considered more robust in case of non-normal distribution [38, 39]. Table 1 shows the baseline sociodemographic statistics of the sample. The final total sample consisted

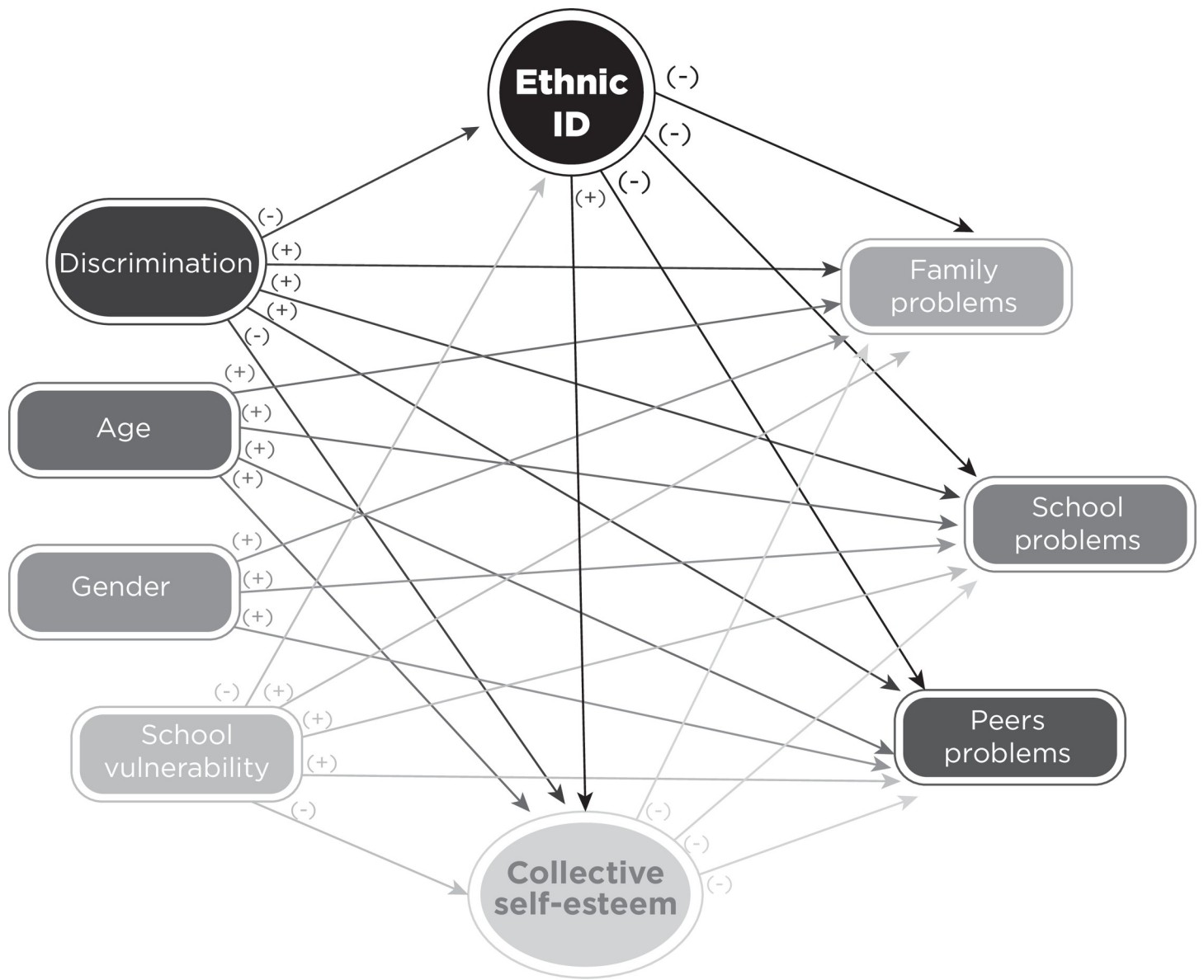

**Fig 1. Theoretical contextual problems model.** The signs (+) or (-) indicate the direction of expected relationship between the variables.

of 3700 students, 1388 belonging to primary and 3212 to secondary schools, respectively. 48.4% were men, and the other 51.6% were women. In the primary group, the mean age was 10 years, and the standard deviation of 0.8. In the secondary group, the mean age was 14.4, and the standard deviation of 1.8. Through data analysis, we found that 91.9% of the students were Chilean, 3,7% Bolivian, 2.8% Peruvian, and the rest were from other nationalities. The ethnic groups with which the students identified themselves were: 54.4% Latin American, 25.9% Aymara, 5,2% Mapuche, 3% Afro-descendant, and the rest were distributed in other ethnic groups. The distribution by grade/course of the schools was quite homogeneous. This study included 39,8% of public schools, 55,5% of subsidized schools, and 4,7% of private schools. Public schools are fully funded by the government. While government-subsidized

**WITHIN**

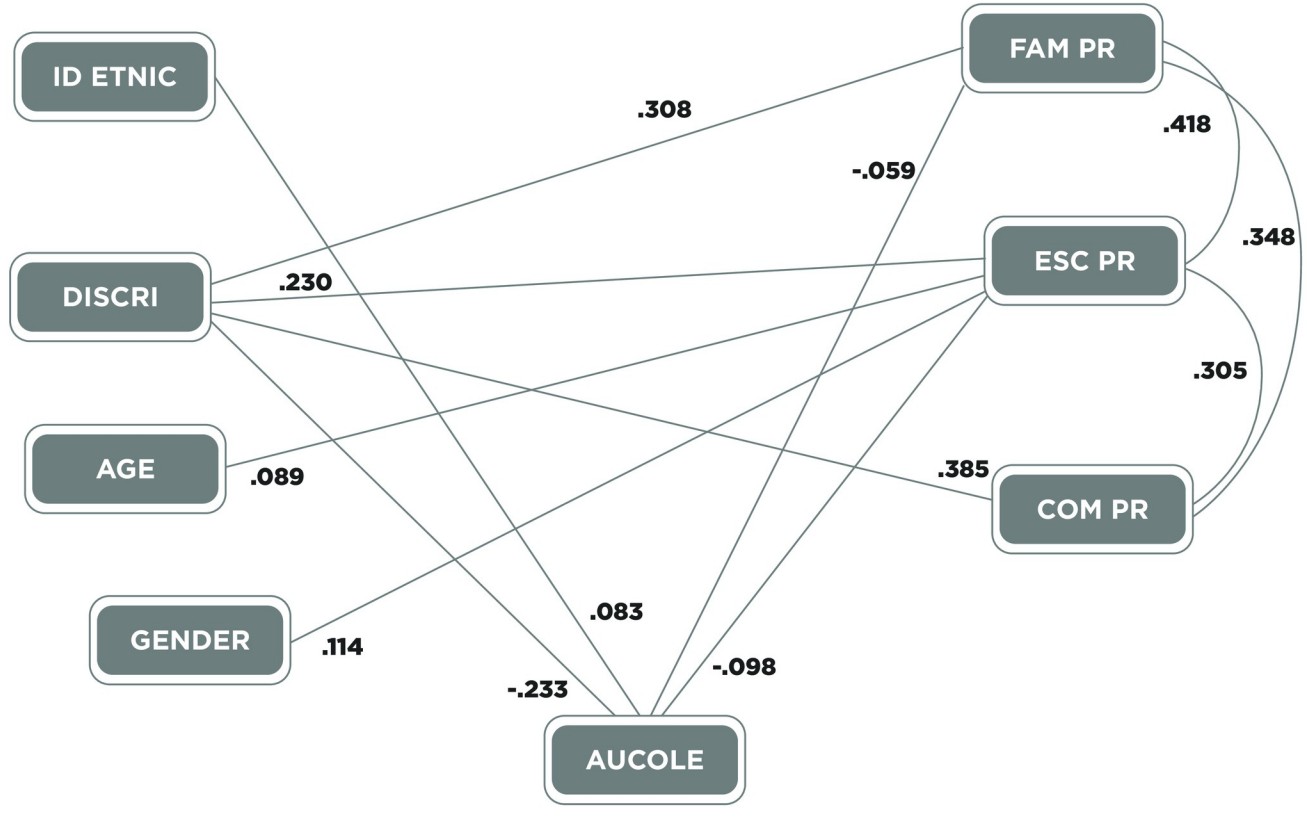

**BETWEEN**

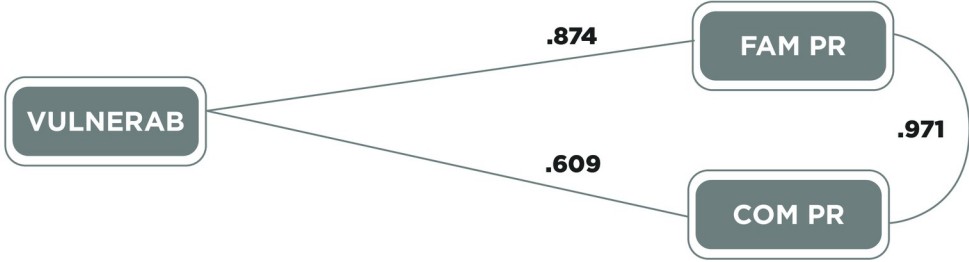

**Fig 2. Primary Path analysis model of contextual problems with their standardized values.** Identic = ethnic identification. Discri = perceived discrimination. Aucole = collective self-esteem. FAM PR = family problems. ESC PR = school problems. COM PR = peer problems. Vulnerab = School vulnerability.

schools only receive partial funding to operate, they are free to charge school fees per pupil. Private schools do not receive any funding, they charge school fees per pupil.

### Instruments

**Everyday Discrimination Scale (EDS) [40].** This is a self-report instrument with one-dimensional structure, and measures discrimination in everyday life. This scale asks how many times subjects have felt discriminated against in their lives. It is a rating scale that is

**Table 1. Sociodemographic characteristics of participants at baseline.**

| Variable | n | % |
|---|---:|---:|
| Gender | | |
| Female | 1908 | 51.6 |
| Male | 1792 | 48.4 |
| Grade | | |
| 4 | 532 | 14.4 |
| 5 | 469 | 12.7 |
| 6 | 387 | 10.5 |
| 7 | 501 | 13.5 |
| 8 | 460 | 12.4 |
| 9 | 407 | 11.0 |
| 10 | 358 | 9.7 |
| 11 | 325 | 8.8 |
| 12 | 261 | 7.1 |
| Ethnicity | | |
| Aymara | 958 | 25.9 |
| Quechua | 38 | 1.0 |
| Mapuche | 192 | 5.2 |
| Afro-descendant | 110 | 3.0 |
| Latin American | 2011 | 54.4 |
| Other | 391 | 10.6 |
| Nationality | | |
| Chile | 3401 | 91.9 |
| Perú | 103 | 2.8 |
| Colombia | 18 | 0.5 |
| Venezuela | 13 | 0.4 |
| Bolivia | 136 | 3.7 |
| Ecuador | 5 | 0.1 |
| Argentina | 4 | 0.1 |
| Other | 20 | 0.5 |
| Type of school | | |
| Public | 1473 | 39.8 |
| Government-subsidized | 2054 | 55.5 |
| Private | 173 | 4.7 |

composed of 6 alternatives ranging from 1 (never) to 6 (almost every day). An example is: "You were treated with less respect than other people." It has acceptable construct validity and adequate evidence of reliability, varying the alpha between 0.8 and 0.9. The Spanish version was used, which had previously been translated into Spanish by the original authors, and has been used in Latino adolescents in other studies in the U.S. [41]. Omega value was 0.88, both in primary and secondary school, in the present study. There is validation in Chile for university students, but not in adolescents. Data analysis was performed with the notion that a higher score meant a higher perceived discrimination.

**Collective Self-Esteem Scale [27].** This instrument consists of 16 items grouped into four subscales, and each subscale consists of four items that are supposed to measure an individual's collective self-esteem. It is a self-report instrument. The four subscales are Membership Esteem, which evaluates the satisfaction with the roles that the individual occupies in the endo-group (i.e., immigrant group); Private Self-esteem, which evaluates the feeling of

achievement and satisfaction that the individual obtains from his/her social belonging to the endo-group; Public Self-esteem, which is linked to the social validation of belonging, that is, how the endo-group is envisioned by the exo-group (i.e., society in general); and Importance to Identity, which is related to the level of importance that the endo-group places on the individual's self-concept. It is a rating scale ranging from 1 (totally disagree) to 7 (totally agree). Data analysis was performed with the notion that a higher score meant higher collective self-esteem. This scale presented adequate evidence of reliability, with an alpha score of 0.76. In the present study, only Private Self-esteem and Importance to Identity dimensions were used, as it was regarded that the other two dimensions are not essential for this study's purposes since the questions were adjusted in relation to nationality. An example of wording is: "I feel proud of my "Colombian/Peruvian/Chilean/. . ." roots". The Spanish version of this instrument was used, which had previously been translated in another study performed in Chile [42], but it has no validation in the children and adolescents population. The omega value was 0.71 in the primary group, and 0.88 in the secondary group, in the present study.

**Contextual sub-scales of family, school, and peer problems from Child and Adolescent Assessment System (SENA) [43].** Developed by Spanish specialists in psychopathology and psychological evaluation, its purpose is to help in the detection of a wide range of emotional and behavioral problems, and it is supposed to be used with children and adolescents ranging from 3 to 18 years old. It is a rating scale and the answers to each item vary from 1 to 5 in the response options of each item, from "never or almost never" to "always or almost always".

There are different versions of this instrument, according to whether the informant is the family, the school, or the student himself, as is the case in the present study. The total of each dimension is the average of the answers that constitute it and can vary between 1 and 5 continuously. It is worth noting that it has been built and validated entirely in Spanish. The results that can be obtained through its application are:

○ Internalized problems: Depression, Anxiety, Social Anxiety, Somatic Complaints, Post-traumatic Symptomatology, and Obsession-Compulsion.

○ Externalized Problems: Attention Problems, Anger Control Problems, Aggressive, Defiant, Hyperactive, and Antisocial behavior.

○ Contextual problems: Family problems, School problems, and Peer problems.

○ Specific Problems: Substance Use, Eating Behavior Problems, Developmental Delay, Learning Problems, Schizotypal Disorder, or Unusual Behavior.

Recently, a study has found that the Cronbach reliability of their subscales is above 0.7 in Spain [44]. In the current research, the three subscales of contextual problems have been used, as well as the general scale of contextual problems, composed of the three subscales. In the 8–12 age version, the sub-scale for family problems has 8 items, the sub-scale for school problems has 7 items, and the sub-scale for peer problems has 7 items, with a total of 22 items measuring contextual problems. In the 12–18 age version, the sub-scale of peer problems has one more item, with the number in the other two sub-scales remaining the same, being 23 items in a total of contextual problems in this version. An example of the wording, in both versions, is: "I have problems at home" (Family problems). SENA has no validation in the infant and juvenile population in Chile. In the present study, the omega values were 0.78 for family problems, 0.76 for school problems, and 0.81 for peer problems in the primary group (8–12 age version). In the secondary group (12–18 age version), the omega values were 0.79 for family problems, 0.73 for school problems, and 0.75 for peer problems.

**Multigroup Ethnic Identity Measure-Revised (MEIM-R) [45].** This self-report instrument is supposed to provide an adequate measure of an individual's ethnic identification. Its revised version consists of 6 items that are scored on a rating scale ranging from 1 (Totally disagree) to 5 (Totally agree). It has two dimensions: Exploration and Commitment. Previous studies have highlighted that instruments based on MEIM, in all its versions, are those that yield more significant results because it relates ethnic identity with other outcome variables, such as well-being or symptomatology [46]. An example of wording is: "I have a strong sense of belonging to my own ethnic group." The omega value was 0.88 in the primary group and 0.92 for the secondary group in the present study. It has validation in the infant and juvenile population in Chile [47].

**Vulnerability index.** Instead of the division between public and non-public schools, a dichotomous criterion of school vulnerability was used. The division between public and non-public schools does not necessarily reflect the vulnerability of the school context, since some non-public schools are free for students.

The classification in high and low vulnerability was carried out according to the statistics presented in the Annual Municipal Development Plan of Arica [48], based on the vulnerability index [49] over three years (2014–2017). An average of 86 per cent of public schools are vulnerable, which is over 77 per cent of the commune. On the other hand, in the same period of time, the vulnerability index of government-subsidized and private schools reached an average of 74% of vulnerability being below the communal percentage. The cut-off point was considered to be above or below average. The highest low vulnerability score is 73 and the lowest high vulnerability score is 78

## Procedure

1. Approval of the ethics committee of the University of Tarapacá. This study is part of a larger Educational Justice Center project.

2. A total of 42 schools in the city of Arica were invited to participate in the study. In total, 29 schools (69%) agreed to participate in the study.

3. It was agreed with the principals of each school to have a 5-minute space at the regular parent-teacher meetings to invite them to participate in the study. Parents who did not attend the meetings received the invitation to participate through a communication from the headteachers of each course.

4. The study's aims and implications were duly explained to each parent, and they were then asked to give their written informed consent. Subsequently, the students themselves were provided with explanations regarding the study's aims and implications and then asked to give their written informed consent.

5. Each questionnaire was applied and finished within 45 minutes, and at least two trained interviewers were present in each room to answer questions regarding its content. At least one teacher of the respective establishment also accompanied the application in each classroom.

## Data analysis

First, analyses were made regarding the sociodemographic descriptions of the sample, and the basic statistics of each variable in both groups. Then Pearson's correlations were calculated for principal variables in each group to establish a significative relationship between variables according to the first hypothesis and partially for the others hypothesis since a significant

relationship is needed before analyzing more complex relationships. A multivariate analysis of variance (MANOVA) was conducted to assess the inclusion of gender and age variables in the subsequent structural equation models. Furthermore, path analysis was performed in both groups to explore the mediating role of collective self-esteem and ethnic identification between discriminations and contextual problems considering the contextual problems into three constructs: family, school, and peer. When analyzing goodness-of-fit, by convention, an RMSEA under .08 is acceptable, and under .05 is optimal, while a CFI and TLI higher than .9 are acceptable, and greater than .95 are optimal. SRMR is good below 0.8. According to previous studies, the CMIN ($\chi$2/gl) is sensitive to sample sizes that go over 200 subjects/participants, although scores under 5 are considered acceptable even with classic criteria [50, 51]. Robust maximum likelihood (MLR) it was the estimator used in path analysis, since it is robust even with not-normal distribution. All scales had enough reliability in the presents study. In general, the alpha and omega values are considered adequate over .7 [52].

In this study, data were analyzed using the SPSS version 21.0 and the Mplus version 8.

The perceived discrimination scale was the last one in the survey, which may have affected the response rate. In total scores of perceived discrimination, in primary grades had 45% missing values and in secondary grades 36%. Although, it is also possible that some students did not want to answer about the discrimination they have experienced. The rest of the items had less than 1% of missing data. Multiple imputation (MI) was used to deal with missing values, and all variables in the database were considered in the imputation process; seven imputed data sets were generated in each group sample. In MI the goodness of fit is calculated in each imputed group, and the final result of each index is the average of all groups. This procedure does not allow for confidence intervals, but it does give the standard deviation. The Bootstrap option is not available for analyzing indirect effects in MI.

Outliers were identified in both groups, considering a z-score equal or greater than ± 3. In primary school, 89 cases (6,4%) were found, and 138 cases (6%) in secondary school. These were not removed from the sample, but the difference between including them or not in the analyses is reported in the results section. The analysis was also considered only with the full cases to see if there were any important differences with the results through imputation.

## Results

### Descriptive statistics and correlations

The asymmetry and kurtosis of both groups are presented in Table 2. It can be seen that the variable with the greatest skewness is peer problems. According to Ryu [53] it is acceptable up to 2 for the asymmetry and 7 for the kurtosis to consider using without problems the analysis that assumed the normal distribution.

Furthermore, Table 3 details the values of the means, standard deviations, and correlations of all variables in the study in both groups. It is observed that perceived discrimination, ethnic identification, and collective self-esteem correlate both with the separate indices of contextual problems. The exceptions in both groups are the non-significant correlations between ethnic identification and peer problems, and between ethnic identification and perceived discrimination, while its correlation with the overall index of contextual problems is the lowest of those that are significant.

### Multivariate analysis of variance

A multivariate analysis of variance (MANOVA) was conducted considering gender and age as independent variables in both groups. The dependent variables were collective self-esteem. discrimination. ethnic identification. contextual problems. problems with the family. problems

**Table 2. Asymmetry and kurtosis.**

|  | Primary | | Secondary | |
|---|---|---|---|---|
| Variable | Asymmetry | Kurtosis | Asymmetry | Kurtosis |
| AUCOLEC | -.898 | 1.085 | -1.003 | .593 |
| DISCRI | 1.286 | .822 | 1.175 | 1.099 |
| ID_ETNIC | -.045 | -.918 | .367 | -.729 |
| FAM PR | 1.661 | 3.087 | 1.092 | .977 |
| ESC PR | 1.339 | 1.703 | 1.076 | 1.046 |
| COM PR | 2.061 | 5.049 | 2.376 | 7.443 |

AUCOLEC = collective self-esteem. DISCRI = perceived discrimination. ID_ETNIC = ethnic identification. FAM PR = family problems. ESC PR = school problems. COM PR = peer problems.

with school and problems with peer. Tables 4 and 5 show the significance of the main effects, and interaction, and the marginal means, respectively, for both groups. The box test is significant in both groups; therefore, the Pillai trace was considered in the global models.

In the primary group, the Pillai trace value was significant for the main effects. being found for sex of $F_{(7,1394)} = 7.685$. $p < .001$. $f^2 = .07$. and for age of $F_{(14,2790)} = 3.403$. $p < .001$. $f^2 = .02$. No significant effect was found on the interaction between gender and age being $F_{(42,13710)} = 1.516$. $p = .212$. $f^2 = .003$. Boys presented greater perceived discrimination, contextual problems in general, as well as problems with school and peer. In ethnic identification

**Table 3. Pearson correlations: Means, standard deviations and correlations of study variables.**

| Primary |  | ACOLECT | DISCRIM | ID_ETNIC | AGE | GENDER | VULNERAB | PD_FAM | PD_ESC | PD_COM | *M* | *SD* |
|---|---|---|---|---|---|---|---|---|---|---|---|---|
|  | AUCOLEC | 1 |  |  |  |  |  |  |  |  | 32,32 | 6,17 |
|  | DISCRI | -,243** | 1 |  |  |  |  |  |  |  | 18,50 | 11,09 |
|  | ID_ETNIC | ,092** | -,038 | 1 |  |  |  |  |  |  | 16,93 | 6,82 |
|  | AGE | ,029 | ,020 | -,063* | 1 |  |  |  |  |  | 9,98 | 0,79 |
|  | GENDER | -,038 | ,188** | -,034 | ,026 | 1 |  |  |  |  |  |  |
|  | VULNERAB | -,111** | ,110** | ,053* | -,001 | ,009 | 1 |  |  |  |  |  |
|  | FAM PR | -,128** | ,347** | -,057* | ,016 | ,072** | ,126** | 1 |  |  | 1,56 | 0,62 |
|  | ESC PR | -,152** | ,283** | -,087** | ,102** | ,168** | ,022 | ,469** | 1 |  | 1,69 | 0,66 |
|  | COM PR | -,100** | ,378** | -,031 | ,042 | ,091** | ,126** | ,444** | ,382** | 1 | 1,46 | 0,62 |
| Secondary |  |  |  |  |  |  |  |  |  |  |  |  |
|  |  | ACOLECT | DISCRIM | ID_ETNIC | AGE | GENDER | VULNERAB | P_FAM | P_ESC | P_COM | *M* | *SD* |
|  | AUCOLEC | 1 |  |  |  |  |  |  |  |  | 33,89 | 5,97 |
|  | DISCRI | -,199** | 1 |  |  |  |  |  |  |  | 18,80 | 9,19 |
|  | ID_ETNIC | ,156** | ,007 | 1 |  |  |  |  |  |  | 14,30 | 6,43 |
|  | AGE | -,160** | ,023 | -,045* | 1 |  |  |  |  |  | 14,35 | 1,77 |
|  | GENDER | ,031 | ,101** | -,056** | ,057** | 1 |  |  |  |  |  |  |
|  | VULNERAB | ,026 | -,021 | ,062** | ,029 | ,014 | 1 |  |  |  |  |  |
|  | FAM PR | -,155** | ,312** | -,066** | -,012 | -,080** | ,066** | 1 |  |  | 1,75 | 0,68 |
|  | ESC PR | -,113** | ,314** | -,156** | ,009 | ,092** | -,051* | ,373** | 1 |  | 1,84 | 0,68 |
|  | COM PR | -,077** | ,397** | ,007 | -,054** | ,118** | ,009 | ,315** | ,387** | 1 | 1,34 | 0,48 |

AUCOLEC = collective self-esteem. DISCRI = perceived discrimination. ID_ETNIC = ethnic identification. FAM PR = family problems. ESC PR = school problems. COM PR = peer problems.

* p < 0,05 (bilateral).

** p < 0,01 (bilateral).

**Table 4. Resume of significant effects.**

| | Primary | | | Secondary | | |
|---|---|---|---|---|---|---|
| Variable | Gender | Age | Gender * Age | Gender | Age | Gender * Age |
| AUCOLEC | .062 | .325 | .910 | .020* | .000** | .015* |
| DISCRI | .000** | .661 | .398 | .003** | .209 | .893 |
| ID_ETNIC | .239 | .012** | .058 | .028* | .753 | .310 |
| FAM PR | .058 | .526 | .100 | .067 | .287 | .494 |
| ESC PR | .000** | .113 | .018** | .000** | .492 | .935 |
| COM PR | .012** | .267 | .785 | .000** | .569 | .080 |

Legend AUCOLEC = collective self-esteem. DISCRI = perceived discrimination. ID_ETNIC = ethnic identification. FAM PR = family problems. ESC PR = school problems. COM PR = peer problems.

* $p < 0.05$ (bilateral).

** $p < 0.01$ (bilateral). Note: interaction effect in primary it was not explained since the global model was not significant.

the age is significant. being found in the DMS post-hoc tests that the 11-year-old group is different from the other two.

In secondary, the Pillai Trace Value was significant for the main effects. being found a value for gender of $F(7.2280) = 6.610$. $p < .001$. $f^2 = .044$. and for age of $F(42.13710) = 5.529$. $p < .001$. $f^2 = .01$. A significant effect was also found for the interaction between gender and age. $F(42.13710) = 1.516$. $p < .05$. $f^2 = .001$.

The gender differences are in collective self-esteem, perceived discrimination, ethnic identification, school problems, and peer problems, with ethnic identification being the only one greater in girls, and the rest are all greater in boys. With respect to age, significant differences were found in collective self-esteem. Post-hoc Gabriel tests found that the 12-year-old group differed significantly in collective self-esteem from the 14. 15. 16 and 17-year-old groups, $p < .000$. The 13-year old group differs from the 15-year old group, $p < .05$, and the 17-year old group, $p < .000$, decreasing in all of them as age increases. An interaction effect between gender and age was obtained. $p < .05$, finding that at 16 years of age, the highest value of collective self-esteem is produced in boys and the lowest in girls.

## Path analysis model of contextual problems

Since no significant correlation between discrimination and ethnic identification was obtained in any sample, the initial theoretical models had to be adjusted before performing a path

**Table 5. Marginal means of MANOVA.**

| | Primary | | | | | Secondary | | | | | | | | |
|---|---|---|---|---|---|---|---|---|---|---|---|---|---|---|
| | Gender | | Age | | | Gender | | Age | | | | | | |
| Variable | Boys | Girls | 9 | 10 | 11 | Boys | Girls | 12 | 13 | 14 | 15 | 16 | 17 | 18 |
| AUCOLEC | 31.9 | 32.8 | 32.5 | 32.0 | 32.8 | 34.3 | 33.5 | 35.7 | 34.9 | 33.6 | 33.1 | 33.5 | 32.2 | 34.3 |
| DISCRI | 20.8 | 16.6 | 17.8 | 19.2 | 18.4 | 19.7 | 17.8 | 17.8 | 18.7 | 19.1 | 19.6 | 19.2 | 18.7 | 16.9 |
| ID_ETNIC | 16.6 | 17.2 | 17.3 | 17.5 | 15.8 | 14.0 | 15.0 | 15.1 | 14.4 | 14.4 | 14.3 | 14.3 | 14.2 | 14.9 |
| FAM PR | 1.5 | 1.5 | 1.5 | 1.5 | 1.5 | 1.7 | 1.7 | 1.7 | 1.7 | 1.7 | 1.7 | 1.7 | 1.7 | 1.5 |
| ESC PR | 1.8 | 1.5 | 1.6 | 1.6 | 1.7 | 1.9 | 1.7 | 1.7 | 1.8 | 1.8 | 1.8 | 1.8 | 1.8 | 1.7 |
| COM PR | 1.5 | 1.4 | 1.3 | 1.4 | 1.4 | 1.4 | 1.3 | 1.4 | 1.3 | 1.3 | 1.3 | 1.4 | 1.3 | 1.2 |

AUCOLEC = collective self-esteem. DISCRI = perceived discrimination. ID_ETNIC = ethnic identification. FAM PR = family problems. ESC PR = school problems. COM PR = peer problems.

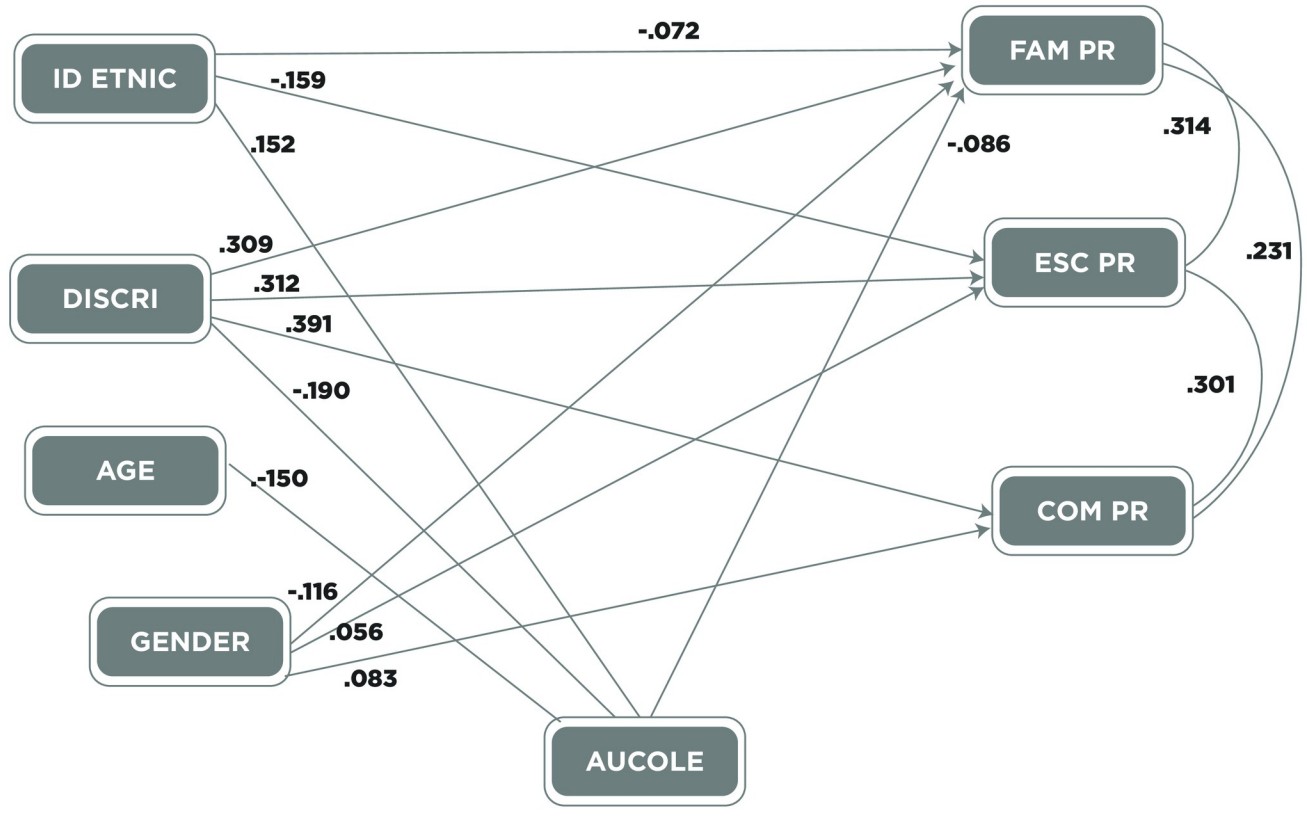

**WITHIN**

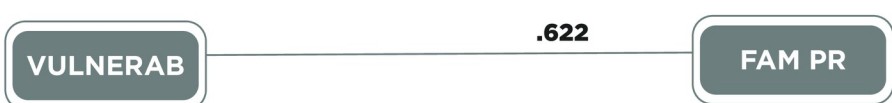

**BETWEEN**

**Fig 3. Secondary Path analysis model of contextual problems with their standardized values.** Identic = ethnic identification. Discri = perceived discrimination. Aucole = collective self-esteem. FAM PR = family problems. ESC PR = school problems. COM PR = peer problems. Vulnerab = School vulnerability.

analysis. Ethnic identification came to be considered as an exogenous variable. Vulnerability was considered according to the correlations obtained with each specific contextual problem in both samples.

**Model in the primary group.** Fig 2 shows the relationships between variables in the primary school sample. It can be seen that in the first level (within) ethnic identification had a direct relationship with collective self-esteem ($b = .083$, $SE = .04$, $p < .001$). Thus, greater ethnic identification was directly associated with greater collective self-esteem. There was only an indirect negative relationship between ethnic identification and school problems, and it was found that greater ethnic identification was related to fewer school problems ($b = -.008$, $SE = .003$, $p < .01$).

Perceived discrimination was directly and positively related to each of the contextual problems. A higher score in perceived discrimination was directly related to greater family problems ($b$ = .308, $SE$ = .04, $p$< .001). A higher score in perceived discrimination was also directly related to a higher score in school problems ($b$ = .230, $SE$ = .04, $p$ < .001), and there was also an indirect positive relationship with the same variable ($b$ = .023, $SE$ = .007, $p$ < .01). Likewise, a higher score of perceived discrimination is related to greater peers' problems ($b$ = .385, $SE$ = .03, $p$ < .001).

Age was directly related only to school problems, so that higher age was significantly related to greater school problems ($b$ = .089, $SE$ = .02, $p$ < .001).

Gender was related only to school problems, with a direct relationship between being male and having greater school problems ($b$ = .114, $SE$ = .03, $p$ < .001).

Collective self-esteem had a direct negative relationship with family problems ($b$ = -.059, $SE$ = .03, $p$ < .05), finding that a higher score in collective self-esteem was related to fewer family problems. Likewise, it presented a direct negative relationship with school problems, being found that higher scores in collective self-esteem are directly related to fewer school problems ($b$ = -.098, $SE$ = .03, $p$ < .001).

At the second level (between), a direct positive relationship was found between school vulnerability and family problems, so that schools with high vulnerability have greater family problems ($b$ = .874, $SE$ = .13, $p$ < .001). A direct positive relationship was also found between vulnerability and problems with classmates, with the most vulnerable schools having the highest scores in peers' problems ($b$ = .609, $SE$ = .19, $p$ < .001). Regarding this analysis, goodness-of-fit indicators are optimal, as demonstrated herein: χ2/gl = 1.23, CFI = .998, TLI = .995, RMSEA = .011 (SD = .006), AIC = 59280.553, BIC = 59641.811, SRMR within = 0.020, SRMR between = 0.020.

**Model in the secondary group.** Fig 3 presents the relationships obtained in the secondary school sample. Ethnic identification was significantly related to family problems. Thus, a higher score of ethnic identification is directly related to lower family problems (b = -.072, SE = .03, p < .001). An indirect relationship between these variables was also found (b = -.013, SE = .004, p < .001). Likewise, a direct relationship was obtained between ethnic identification and school problems, so that greater ethnic identification is inversely related to school problems (b = -.159, SE = .03, p < .001). In addition, ethnic identification was significantly related to collective self-esteem, and it was observed that a higher score in ethnic identification was related to higher collective self-esteem (b = .152, SE = .02, p < .001).

Discrimination was again associated with each of the three specific contextual problems. A higher score in perceived discrimination was related to greater family problems (b = .309, SE = .03, p < .001). It was also found that a higher score in perceived discrimination was related to greater school problems (b = .312, SE = .03, p < .001). In addition, a higher score on perceived discrimination was related to greater peers' problems (b = .391, SE = .03, p < .001).

Age only obtained a significant relationship with collective self-esteem, so a higher age was associated with lower collective self-esteem (b = -.150, SE = .02, p < .001). There was also an indirect relationship between these variables (b = -.013, SE = .004, p < .001).

Gender was directly related to family problems, so being female was associated with greater family problems (b = -.116, SE = .02, p < .001). Gender was also directly related to school problems, so being male was associated with greater school problems (b = .056, SE = .02, p < .01). In addition, gender was directly related to peers' problems, so being male was associated with having greater peers' problems (b = .083, SE = .02, p < .001).

Collective self-esteem was only directly related to family problems, so higher collective self-esteem was associated with fewer family problems (b = -.086, SE = .02, p < .001). An indirect relationship was also found between both variables (b = .013, SE = .004, p < .001).

At a second level (between), vulnerability of the school context was significantly associated with family problems. Thus, the schools with high vulnerability had greater family problems (b = .622, SE = .19, p < .001). Regarding this analysis, goodness-of-fit indicators are optimal, as demonstrated herein: $\chi2/gl$ = 3.43, CFI = .989, TLI = .964, RMSEA = .032 (SD = .002), SRMR within = 0.019, SRMR between = 0.005.

Goodness-of-fit for all models are optimal since all parameters are beyond the acceptable ranges, and some even beyond the optimal ranges. Finally, collective self-esteem was regarded with a role of partial mediation for both ethnic identification and perceived discrimination in all models.

### Effect to removes outliers

The results of removing the outliers before multiple imputation should be considered as reference. In the primary model, all relationships are maintained. The maximum effect variation in all relationships is around .04, except for the effect of perceived discrimination on school problems, which decreases by .06. In the model for secondary schools, all relationships are maintained, and the maximum variation in the effects on relationships is .04. In both Path analysis models, optimum goodness fit indicators are maintained.

### Effect of considering only full cases

When only full cases were kept without imputation, they were only analyzed using list-wise deletion and all significant relationships were maintained in both models. In the primary school model, the maximum variation in all relationships was around .03 at the within level, and .05 at the between level. In the secondary schools' model, the maximum variation in the effects on relationships was .03 at the within level, and .003 at the between level. In both Path analysis models, optimum goodness fit indicators were maintained.

## Discussion

The aim of this study was divided into two sections: first. to establish the relationship between perceived discrimination and contextual problems in inter-ethnic students aged 9 to 19 living in the northern region of Chile; and second, to identify the role that ethnic identification and collective self-esteem play within this relation. The three hypotheses that derived from this objective were: 1) The greater the perceived discrimination, the greater the contextual problems. 2) Ethnic identification has a mediating role between perceived discrimination and contextual problems. 3) Collective self-esteem has a mediating role between perceived discrimination and contextual problems.

### Relation between perceived discrimination and contextual problems

Evidence was found to fully support the first hypothesis in both groups. The results indicate that perceived discrimination has a direct relationship with contextual problems, so it can be inferred that exacerbated discriminatory behaviors towards students may increase their family, school, and peer problems. Further, it can be noted that peer problems was the most directly related area with perceived discrimination. In opposition to this finding, early studies on the ecological model gave more importance to students' school relations with their teachers/school than with their peer [21. 22]. However, since our findings demonstrate that peer problems was the most directly related area with students discrimination experience, it seems necessary to give importance to peer relationship. Generally, the theory pays more attention to the relationship between students and teachers than to the relationship between students and their peers.

Nevertheless, our results coincide with recent research on children and adolescent's development. These studies consider peer relations equally as important as family and teachers relations [24. 25].

In addition, there are newer propositions regarding the ecological model which stated that humans develop themselves through progressive complex reciprocal interactions, which involve an individual (in this case the student) and the objects, symbols and others individuals in his/her external environment (in this case family, school and peer), These interactions impact the individual's development. These interactions must be regular and placed over extended periods of time; this new concept was called proximal process [54]. Thus, based on the concepts of proximal processes, if we consider that peer spend much more time with the student than their parents or teachers/school, it is quite logical to assume that their role is more relevant than the original theory stated and, therefore, it seems relevant to pay attention to the effects of perceived discrimination on peer problems when analyzing children and adolescent development.

Moreover, problems in peer relations are generally studied within the bullying perspective [55]. However, this approach is limited since there are other conflicts that do not necessarily involve students being bullied and still involve discrimination experience. Thus, further investigations on these relations are needed since they are relevant for children's and adolescents' development and could be central to this matter.

The direct significant relationship obtained in this study between perceived discrimination and family problems is consistent with previous research [11. 19], although they used different instruments to measure family problems. These findings are also in line with the results obtained by a study conducted with an adult Latin American population [18].

As for school problems, a review study noted that previous studies have usually investigated the student-teacher relationship in the context of teacher discriminatory behavior or in aspects not directly related to conflict, such as school climate investigations; and that previous studies have usually investigated the direct student-school relation in the contexts of size, culture, and safety of the school for the students [23]. Thus, since this study's findings indicate that school problems are also related with discrimination. it seems necessary for further studies to investigate school problems, as it may further improve the understanding of children's and adolescents' development.

It should be noted that in models, the direct relationship of discrimination with school problems increases by .08 or more, but almost did not change in family and peer problems. It is possible to hypothesize that secondary school students have experienced a cumulative effect of perceived discrimination, and this increases the direct effect of discrimination on contextual problems. Future research may address this possibility.

## Ethnic identification and collective self-esteem role

The second hypothesis was not supported by evidence since ethnic identification was not directly related with perceived discrimination in any group. Furthermore, in the models of both groups, it was not related to all specific contextual problems, but to some. The present results rule out the mediation of ethnic identification in the relation between perceived discrimination and contextual problems, although it may have presented a direct relationship with some contextual problems. In secondary groups, ethnic identification has a direct relationship with school problems, but also with family problems. These results are supported by research which stated that perceived discrimination is not significantly related to the ethnic identification of adolescents, although they may have used different instruments to measure discrimination [56, 57].

However, one of these previous studies has brought up another variable into this question: maybe ethnic identification has not matured enough in adolescents, so it would not establish such a relation with discrimination due to it still being in development [56]. In partial corroboration to that notion, another study performed on Latin American adults was able to effectively find a relation between both variables (ethnic identification and perceived discrimination) [58]. On the other hand, another study performed on Asian American adults was not able to effectively find a relation between both variables [59].

Notwithstanding, some of these results partially contradict what a previous study found using the same MEIM-R scale that was used in this study, as a relation between perceived discrimination and ethnic identification in adolescents was found [60]. In addition, other studies that did not necessarily use this scale have also found solid relations between both variables in adolescents [28, 31, 61]. In short, there are mixed results that do not allow to either discard or affirm this relation, and this has been previously highlighted in a previous meta-analysis study [46].

Furthermore, the direct relationship of ethnic identification with family problems found in this research contradicts the findings of previous research, which stated that ethnic identification significantly affected parent-child relations in African Americans, but not in Latin Americans [33].

Evidence was found that partially supports the third hypothesis of the study, which affirms the collective self-esteem mediation relationship between discrimination and contextual problems. In the models it does not appear completely, because collective self-esteem affects two contextual problems only in the primary school group. In the secondary group, it affects exclusively family problems.

Moreover, the negative effect of discrimination on collective self-esteem is expected and is consistent with previous research conducted with adolescents [26. 28], as well as those conducted with children [62]. However, in another study, a relation between the two variables was not found in children, but a different instrument was used to measure discrimination, and that should be taken into account when comparing both [63].

Additionally, the relation between ethnic identification and collective self-esteem coincides with the results of a previous study which found a positive relation between the two [28] and another study that found that ethnic identification predicts collective self-esteem [37]. Furthermore, it is logical to situate both variables within the macrosystem of Bronfenbrenner's theory since they form part of the culture in which the individual is immersed.

Further, this research had an adequate treatment of missing values–by multiple imputation (MI) in the present study—taking into account the recommendations of previous studies [64]. This same study emphasized that significant statistical effects should not be discarded, even if they were modest, but it is quite clear that the role of mediation of collective self-esteem does not seem sufficient to focus strategized interventions on it. On the other hand, perceived discrimination and ethnic identification seemed as more reasonable focal points when creating strategized intervention efforts, and since both ethnic identification and collective self-esteem are linked to the individual's identity, it seems reasonable to include the measurement of identity when strategizing these interventions.

In terms of strengths, the path analysis models proposed in this study had appropriate goodness-of-fit indicators, and provided relevant findings for this study's aim. Another strength was the wide age range of participants, and this is a relevant aspect because this type of study usually investigates either adolescents or children, not both. Undoubtedly, it was an advantage to use a multilevel analysis, which has allowed to rule out that individual differences are actually due to differences between schools.

Nonetheless, this study had several limitations. In terms of data gathering, convenience sampling was performed, so it reduced the possibilities of result generalization, and further

studies should try to select more specific sample groups to provide more accurate notions regarding the topic of study. Further, only one region of the country was considered and incorporated, so further studies that encompass a more geographically varied sample would be ideal. Moreover, although the path analysis models were guided and performed based on previous theories, they did not offer experimental evidence that allows for the establishment of causality among the variables. Future studies should try to utilize statistical models that may allow for the establishment of causality among the variables to further improve the range of conclusions of the study. Finally, this study also relies exclusively on student's self-reports, so other studies that offer different types of reports and observations in terms of data collection are required to further investigate perceived discrimination among children and adolescents in Chile. Furthermore, although the SENA instrument has an inconsistency scale, it does not have a social desirability scale.

Interestingly, school vulnerability plays a different role in both groups. It was the only variable that remained significant at the between group level. It is remarkable that it was directly related to both, family and school problems in the primary sample, but only with family problems in the secondary sample. This could imply that the decisive role of the vulnerability of the school context is concentrated in primary education.

Finally, regarding future researches, we believe that there are some specific changes regarding the methodological approach that could further contribute to the understanding of the studied phenomenon, and are described herein: including other contextual aspects and problems, such as the children's and adolescents' neighborhood; the inclusion of other variables, such as personal self-esteem in the statistical model; the inclusion of others' perspectives regarding the discriminatory process, such as the parents' and teachers' (an action that would be consistent with the ecological model that was explored in this study); and changing the study's perspective to a longitudinal one.

## Conclusions

This study shows that perceived discrimination is directly related to contextual problems at the family, school and peer levels; ethnic identification has a direct relationship to school, and family problems in secondary students; and collective self-esteem has a mediating role for both ethnic identification and perceived discrimination over contextual problems.

## Supporting information

**S1 File.**
(XLSX)

**S2 File.**
(DOCX)

**S3 File.**
(SAV)

**S4 File.**
(SAV)

## Author Contributions

**Conceptualization:** Jerome Flores, Alejandra Caqueo-Urízar, Lirna Quintana, Alfonso Urzúa, Matías Irarrázaval.

**Data curation:** Jerome Flores, Alejandra Caqueo-Urízar.

**Formal analysis:** Jerome Flores, Alejandra Caqueo-Urízar.

**Investigation:** Jerome Flores, Alejandra Caqueo-Urízar, Lirna Quintana, Alfonso Urzúa, Matías Irarrázaval.

**Methodology:** Jerome Flores, Alejandra Caqueo-Urízar.

**Project administration:** Jerome Flores, Alejandra Caqueo-Urízar.

**Writing – original draft:** Jerome Flores, Alejandra Caqueo-Urízar, Lirna Quintana, Alfonso Urzúa, Matías Irarrázaval.

**Writing – review & editing:** Jerome Flores, Alejandra Caqueo-Urízar, Lirna Quintana, Alfonso Urzúa, Matías Irarrázaval.

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
