## [Decision Letter · Decision Letter 0]

18 May 2020

PONE-D-20-05463

Perceived discrimination and contextual problems among children and adolescents in Northern Chile

PLOS ONE

Dear Jerome Aníbal,

Thank you for submitting your manuscript to PLOS ONE. After careful consideration, we feel that it has merit but does not fully meet PLOS ONE’s publication criteria as it currently stands. Therefore, we invite you to submit a revised version of the manuscript that addresses the points raised during the review process.

We would appreciate receiving your revised manuscript by 16/07/2020. To enhance the reproducibility of your results, we recommend that if applicable you deposit your laboratory protocols in protocols.io, where a protocol can be assigned its own identifier (DOI) such that it can be cited independently in the future. For instructions see: http://journals.plos.org/plosone/s/submission-guidelines#loc-laboratory-protocols

We look forward to receiving your revised manuscript.

Kind regards,

Eduardo Fonseca-Pedrero, PhD

Academic Editor

PLOS ONE

Journal Requirements:

3. Please use periods "." instead of commas"," for decimal points.

4. Please include additional information regarding the survey or questionnaire used in the study and ensure that you have provided sufficient details that others could replicate the analyses. For instance, if you developed a questionnaire as part of this study and it is not under a copyright more restrictive than CC-BY, please include a copy, in both the original language and English, as Supporting Information. Please clarify whether you used a Spanish version of the MEIM-R and if so how it was validated.

Additional Editor Comments (if provided):

The work entitled "Perceived discrimination and contextual problems among children and adolescents in Northern Chile” is a good research paper. However, I have a few comments to make that should be addressed before I recommend this manuscript for publication to PLOS ONE:

1.- Please, add empirical information in the abstract.

2.- Please, add an operative definition of the main constructs used.

3.- In English, a decimals point is used, not comma.

4.- Please, add the main hypothesis at the end of the introduction.

5.- Add more information about sample and sampling procedure as well as others variables like: % by age, socio-economic level, IQ, previous mental disorders, etc. Add all the sample information in the method section, not as a results.

6.- Do you have any information about non-response? Describe inclusion/exclusion criteria if part of the data was excluded from the analysis.

7.- Were outliers removed from the data? Which method did you use to deal with missing data in the analyses? What variables are related to missing data?

8.- Add information about the psychometric properties of the measures used in this study. In addition, add Omega values, not Cronbach. Cronbach alpha has a lot of limitations from a psychometric point of view.

9.- Add subheadings in the results section (as function of the goals).

10.- Add more information about mediation analyses performed.

11.- Please, before to SEM analysis, compute a MANOVA or MANCOVA in mean comparisons by gender and age. Add effect sizes in mean comparisons.

12.- Please, add other godness of fit indices, eg.: TLI, BIC, IC RMSEA 90%, etc. Add information about goodness of fit indices. Test model assumptions (e.g., normality).

13.- Add more limitations (e.g., self-reports, no scale to test social desirability).

14.- Check English grammar.

Reviewers' comments:

Reviewer's Responses to Questions

**Comments to the Author**

1. Is the manuscript technically sound, and do the data support the conclusions?

Reviewer #1: Partly

Reviewer #2: Yes

2. Has the statistical analysis been performed appropriately and rigorously? 

Reviewer #1: Yes

Reviewer #2: Yes

3. Have the authors made all data underlying the findings in their manuscript fully available?

Reviewer #1: Yes

Reviewer #2: Yes

4. Is the manuscript presented in an intelligible fashion and written in standard English?

Reviewer #1: No

Reviewer #2: Yes

5. Review Comments to the Author

Reviewer #1: The work entitled “Perceived discrimination and contextual problems among children and adolescents in Northern Chile" is of great interest. The research is very stimulating; it contains new scientific knowledge and provides comprehensive information for further development of this productive line of research. However, I have some comments to make that should be addressed before I recommend this manuscript for publication.

- In the introduction, authors should consider devoting some lines about psychological problems in children and adolescence. With this regard recent relevant research analyzing prevalence of emotional and behavioural problems at those ages should be introduced (Dalsgaard et al., 2020; Ortuño-Sierra, Aritio-Solana, & Fonseca-Pedrero, 2018).

- Authors should consider the following affirmation in the introduction: “Based on the aforementioned facts, we believe this study could contribute significantly to reduce the effects of discrimination in schools within the northernmost regions of Chile and other applicable contexts”. First, this sentence would better fit in the discussion section. Second I am not completely sure about the adequacy of such affirmation considering the nature of the study. Authors may contribute relevant information to comprehend the nature of the problema and from that prevent discrimination but they dont know if that reduce significantly the effects of discrimination.

- In addition, the following sentence should be placed in the discussion section. “The aim of the study was achieved, as perceived discrimination and ethnic identity were regarded”. With this regard, the objectives section should provide information about hypothesis.

- In the participants section, authors may consider including the age distribution.

- In the instruments section, authors please check for grammatical mistakes in the following sentence: “It is a Likert-type scale consisted…”

- In addition, authors should described the estimator when talking about evidences of internal consistency of the scores. Also, evidences of the MEIM-R are not provided.

- Moreover, authors should consider explaining whether the instruments are self-reported or information could or have to be provided by parents, teacher and/or clinicals. This is relevant beacuse in the procedure it is not clear (please also consider modifying) if the students themselves completed the questionnaires or if the teachers or someone else provided information about the students.

- With this regard, and assuming the students completed the questionnaires, have the authors consider that there are concerns raised about children under 11-10 years old answering self-reported questionnaires due to the lack of instrospection?

- Also, it seems after eading the procedure that the written consent was asked to the participants. I do not know how appropriate it is that under 18 and in this case children of 8,9,10 years old provide this kind of consent. In my opinion, parents, teachers or legal tutors should have asked for this aspect.

- I would also consider making the objectives more specifics and related to the results and conclusions. It is not till the results and the discussion section when one start realizing why the authors includen the SEM analysis. Please also explain in the data analysis section the reference values for the goodnes-of-fit índices values instead of doing it in the results section.

- Finally, authors should consider revising the writing when talking about internal consistency or validity in some parts of the paper. First of all, validity is not a property of the test but inferences of the scores, and also, there are sources or validity evidences, as it is reflected in the APA standards. Attending to this approach it would be more appropriate to talk about evidences of internal structure or evidences of relation with other variables or external variables. In addition, the reliability is not a characteristic of the test. It is more correct to talk about reliability of the scores or estimation of the reliability of the scores (Prieto & Delgado, 2010).

Reviewer #2: Lines 176, 183, 200, 220: please indicate the metric properties of the instruments and clarify if they have validation in your country. Not only Spanish translation.

Lines 262 to 263: It is not clear why you say that there is no statistically significant correlation between ethnic identity and problems with peers. In the correlation chart the information is different.

6. PLOS authors have the option to publish the peer review history of their article (what does this mean?). If published, this will include your full peer review and any attached files.

Reviewer #1: No

Reviewer #2: No

---

## [Author Response · Author response to Decision Letter 0]

31 Jul 2020

Dear Editor and reviewers:

We are grateful of all comments. We decide to divide the sample in two groups, primary and secondary, since we used different SENA age version in each one. Even if both SENA version share more than 70% items, slight differences in the rest can affect the analysis, and the secondary version has one item more.

1.- Please, add empirical information in the abstract.

It was adding now

2.- Please, add an operative definition of the main constructs used.

There is not a cut-off point for to establish enough collective self-esteem or ethnic identity. Then it was not specified. SENA has cut-off for four risk zones, but it was not used because a continuous variable can show clearer the relationship.

3.- In English, a decimals point is used, not comma.

It was changed

4.- Please, add the main hypothesis at the end of the introduction.

We added three hypotheses

5.- Add more information about sample and sampling procedure as well as others variables like: % by age, socio-economic level, IQ, previous mental disorders, etc. Add all the sample information in the method

section, not as a results.

It was added all information and change the position in the document. We don’t ask socioeconomic level to child’s because the primary group probably don’t know

6.- Do you have any information about non-response? Describe inclusion/exclusion criteria if part of the data was excluded from the analysis.

We added criteria

7.- Were outliers removed from the data? Which method did you use to deal with

missing data in the analyses? What variables are related to missing data?

We identified out-layers in both groups, but we dint remove. We explain the differences between include or not at the end of result section. 

The effects of removing the out-layers are considered. The vast majority of the relationships are maintained and vary in a range of 3-5%. Only one relationship ceases to be significant, ethnic identity on school problems in primary group. In sum, this means that relationships are not generated by the presence of out-layers. In any case, by removing them and recalculating the Z-scores, new out-layers appear. Since the MLR estimator that is resistant to non-normality has been chosen, it is decided to keep all the cases in the sample.

8.- Add information about the psychometric properties of the measures used in this study. In addition, add Omega values, not Cronbach. Cronbach alpha has a lot of limitations from a psychometric point of view.

We added all information about each scale and omega indices for all

9.- Add subheadings in the results section (as function of the goals).

We added subheadings in this section

10.- Add more information about mediation analyses performed.

We added all information

11.- Please, before to SEM analysis, compute a MANOVA or MANCOVA in mean comparisons by gender and age. Add effect sizes in mean comparisons.

We added a MANOVA and effect sizes

12.- Please, add other godness of fit indices, eg.: TLI, BIC, IC RMSEA 90%, etc. Add information about

goodness of fit indices. Test model assumptions (e.g., normality).

We added information about normality, and more fit indices

13.- Add more limitations (e.g., self-reports, no scale to test social desirability).

We added 

14.- Check English grammar.

We reviewed it

Reviewer #1: The work entitled “Perceived discrimination and contextual

problems among children and adolescents in Northern Chile" is of great

interest. The research is very stimulating; it contains new scientific knowledge and

provides comprehensive information for further development of this productive line

of research. However, I have some comments to make that should be addressed before I

recommend this manuscript for publication.

- In the introduction, authors should consider devoting some lines about

psychological problems in children and adolescence. With this regard recent relevant research analyzing prevalence of emotional and behavioural problems at those ages should be introduced (Dalsgaard et al., 2020; Ortuño-Sierra, Aritio-Solana,

& Fonseca-Pedrero, 2018).

The inclusion of mental health studies diverts the focus from discrimination in relation to contextual problems. We have reviewed the articles, but they do not mention the contextual problems as to connect them with our article

- Authors should consider the following affirmation in the introduction:

“Based on the aforementioned facts,

we believe this study could contribute significantly to reduce the effects of

discrimination in schools within the northernmost regions of Chile and other

applicable contexts”. First, this sentence would better fit in the discussion

section. Second I am not completely sure about the adequacy of such affirmation

considering the nature of the study. Authors may contribute relevant information to

comprehend the nature of the problema and from that prevent discrimination but they

dont know if that reduce significantly the effects of discrimination.

We rewording it to keep in the same section as part of relevance to investigate this topic

- In addition, the following sentence should be placed in the discussion section.

“The aim of the study was achieved, as perceived discrimination and ethnic identity were regarded”. With this regard, the objectives section should provide information about hypothesis.

We removed that sentence.

- In the participants section, authors may consider including the age distribution.

We added it

- In the instruments section, authors please check

for grammatical mistakes in the following sentence: “It is a Likert-type scale

consisted…”

We removed it

- In addition, authors should described the estimator when talking about evidences of internal consistency of the scores. Also, evidences of the MEIM-R are not provided.

We described MLR estimator in SEM, and added omega indices in each instrument

- Moreover, authors should consider explaining whether the instruments are

self-reported or information could or have to be provided by parents, teacher and/or

clinicals. This is relevant beacuse in the procedure it is not clear (please also

consider modifying) if the students themselves completed the questionnaires or if

the teachers or someone else provided information about the students.

We added it. Also, we expand procedure section

- With this regard, and assuming the students completed the questionnaires, have the authors consider that there are concerns raised about children under 11-10 years old

answering self-reported questionnaires due to the lack of instrospection?

We explained that there are two SENA instruments. One for each group, specially address to be easy to understand and answer.

- Also, it seems after eading the procedure that

the written consent was asked to the participants. I do not know how appropriate it

is that under 18 and in this case children of 8,9,10 years old provide this kind of

consent. In my opinion, parents, teachers or legal tutors should have asked for this

aspect.

We added information in procedure section. First, we ask to parents. Then we ask student themselves. We think that is more respectful with the child’s and adolescents.

- I would also consider making the objectives more specifics and related to the

results and conclusions. It is not till the results and the discussion section when

one start realizing why the authors includen the SEM analysis. 

We added three hypotheses to connect objective with analysis and results

Please also explain

in the data analysis section the reference values for the goodnes-of-fit

índices values instead of doing it in the results section.

We moved it

- Finally, authors should consider revising the writing when talking about internal

consistency or validity in some parts of the paper. First of all, validity is not a

property of the test but inferences of the scores, and also, there are sources or

validity evidences, as it is reflected in the APA standards. Attending to this

approach it would be more appropriate to talk about evidences of internal structure

or evidences of relation with other variables or external variables. In addition,

the reliability is not a characteristic of the test. It is more correct to talk

about reliability of the scores or estimation of the reliability of the scores

(Prieto & Delgado, 2010).

We rewording it

Reviewer #2: Lines 176, 183, 200, 220: please indicate the metric properties of the

instruments and clarify if they have validation in your country. Not only Spanish

translation.

We added it

Lines 262 to 263: It is not clear why you say that there is no statistically

significant correlation between ethnic identity and problems with peers. In the

correlation chart the information is different.

 It was non-significant correlation. We remarked this again.

r= ,020 

Thanks

---

## [Editor Report · Decision Letter 1]

14 Sep 2020

PONE-D-20-05463R1

Perceived discrimination and contextual problems among children and adolescents in Northern Chile

PLOS ONE

Dear Dr. Flores,

Thank you for submitting your manuscript to PLOS ONE. After careful consideration, we feel that it has merit but does not fully meet PLOS ONE’s publication criteria as it currently stands. Therefore, we invite you to submit a revised version of the manuscript that addresses the points raised during the review process.

A rebuttal letter that responds to each point raised by the academic editor. You should upload this letter as a separate file labeled 'Response to Reviewers'.A marked-up copy of your manuscript that highlights changes made to the original version. You should upload this as a separate file labeled 'Revised Manuscript with Track Changes'.An unmarked version of your revised paper without tracked changes. You should upload this as a separate file labeled 'Manuscript'.

We look forward to receiving your revised manuscript.

Kind regards,

Daniel Romer

Academic Editor

PLOS ONE

Additional Editor Comments (if provided):

Thank you for resubmitting your paper to PLOS ONE. I have now assumed the editorship of your paper and have a number of suggestions to make your paper stronger because it is not acceptable for the journal in its present form.

First, to test your hypotheses, it will be important for you to formulate your SEM more clearly in line with them. You are testing the relation between perceived discrimination and various problems that children have, and this requires that discrimination is the exogenous variable in your model and the two other variables (collective esteem and ethnic identification) are mediators between discrimination and your outcomes. Your models should reflect that. As they are currently presented, you have ethnic identification as exogenous and collective esteem as a mediator. Please redo you models so that both of the other two variables are mediators between discrimination and problems.

Second, please add age and gender to the correlation matrices for both age groups so that one can see how these other exogenous variables are related to your survey measures.

Third, if your correlation matrices are correct, there is a dramatic difference between the younger and the older age groups in the way that discrimination is related to problems. It is negatively related in the younger group but positively in the older youth. This should be a major focus of your analyses and interpretations. I do not see any indication that you have recognized this in any of your models.

Fourth, you have a major problem in regard to clustering by school. Your analyses collapse over schools and therefore it is not clear how much is due to school differences versus individual differences. You can remove school differences before analyzing the data so that you are more certain that the relations you are seeing are due to individual differences and not differences between schools. Or you can include dummy variables for schools in your models so that they are controlled. But you have to do something to handle this problem.

I do not understand what you mean by “unitary model of contextual problems” in line 385. Are you collapsing over all of the problem scales or just using the context outcome?

I think it would be clearer if you called your ethnic identity variable “ethnic identification,” which makes it clearer that it is not actual identity that you are measuring but how strongly youth identify with their identity.

It is not clear how the multiple imputation was involved in the analyses. How did you use that? Was it for the MANOVA? It is not clear how it would be used in the SEM. It is also not clear how you used the effect size measure in lines 315-317. Nor is it clear how you used the missing at random test in lines 318-322. A clearer explanation of how much data were missing would be helpful. With SEM, maximum likelihood imputation is usually used to handle missing data.

Please describe the difference between Public and Subsidized schools. Please provide a reference for the omega statistic. Only scales that use the agree-disagree format are truly Likert scales. Other types of responses are merely rating scales (e.g., never to every day). It is not clear why you say that the self-esteem scale is a self-report instrument (line 217) when all of your assessments involve self-report. It is not clear what you mean in lines 228-229 about “adjusted in relation to nationality.”

You should also remove all statements of causal direction in your description of the findings. Rather than saying a variable had a direct effect, you can say that it was directly related. You are also using regression in your models, so path weights are regression weights and not correlations. Also, please describe how you tested indirect relations. This is usually done with bootstrapping.

Finally, you still have issues with appropriate use of English. I highlighted some places where the English is not clear.

---

## [Author Response · Author response to Decision Letter 1]

2 Oct 2020

Dear Editor and Reviewers:

Thank you for all the helpful comments. We present below our answers to each observation.

First, to test your hypotheses, it will be important for you to formulate your SEM more clearly in line with them. You are testing the relation between perceived discrimination and various problems that children have, and this requires that discrimination is the exogenous variable in your model and the two other variables (collective esteem and ethnic identification) are mediators between discrimination and your outcomes. Your models should reflect that. As they are currently presented, you have ethnic identification as exogenous and collective esteem as a mediator. Please redo you models so that both of the other two variables are mediators between discrimination and problems.

Re: we added two theoretical models in the beginning. The reason these models changes it was along with the data analysis, we noted that there is not a correlation between discrimination and ethnic identity. Then not makes sense to keep the original model. Anyway, if we run the original model, the result is the same because there is no significant relationship between discrimination and ethnic identity, and the arrows in the model only reflect significant relationships. Now we explain the process more clear.

Second, please add age and gender to the correlation matrices for both age groups so that one can see how these other exogenous variables are related to your survey measures.

RE: We can not add sex (binary) in a Pearson correlation matrix. Besides, we already test gender in MANOVA. We prefer to keep gender and age in MANOVA to show any interaction effect. Anyway, there is also a way to show if these variables need to be added to the SEM models.

Third, if your correlation matrices are correct, there is a dramatic difference between the younger and the older age groups in the way that discrimination is related to problems. It is negatively related in the younger group but positively in the older youth. This should be a major focus of your analyses and interpretations. I do not see any indication that you have recognized this in any of your models.

Re: We apologize for that. It was a mistake in primary values. We already correct sings. 

Fourth, you have a major problem in regard to clustering by school. Your analyses collapse over schools and therefore it is not clear how much is due to school differences versus individual differences. You can remove school differences before analyzing the data so that you are more certain that the relations you are seeing are due to individual differences and not differences between schools. Or you can include dummy variables for schools in your models so that they are controlled. But you have to do something to handle this problem.

RE: Thanks for this suggestion. Like three types of school are clearly not equivalent groups (private school are 5% approximate in bot samples, don't makes sense to compare in this way). We have the option to consider in the binary way public (only public) and No public school (subsidized and particular). But these no necessary reflect vulnerability since several subsidized schools have not cost for children. Then we added now a vulnerability school variable in both models, on basis a national vulnerability index. We added index vulnerability information in the instruments section. It is better tan classification by public or not public school. We estimate both (school type and school vulnerability), but the vulnerability criterion was clearly more significant. The models keep similar values and good fit.

I do not understand what you mean by "unitary model of contextual problems" in line 385. Are you collapsing over all of the problem scales or just using the context outcome?

RE: There are three specific contextual problems scales in SENA: family, schools, and peers. Also, all these are combined in one global index of contextual problems. Unitary models are the last case, but now we changed the name to global model along with all manuscript. Specific models are the first case. Now we added two figures of theoretical models to improve clarity. Along with the paper, we try to decide which one is a better model to consider.

I think it would be clearer if you called your ethnic identity variable "ethnic identification," which makes it clearer that it is not actual identity that you are measuring but how strongly youth identify with their identity.

Re: we try to keep the original sense of the scale, but we can add this topic in the discussion 

It is not clear how the multiple imputation was involved in the analyses. How did you use that? 

Re: Mulitple imputations and FIML are asymptotically equivalent. We chose to use MI because is robust in the case to non-normal distribution according to Nakagawa (2015) and Enders(2010). all variables were used in each sample to perform multiple imputation of missing values

Was it for the MANOVA? It is not clear how it would be used in the SEM. 

RE: In the previous review, MANOVA was requested to establish whether age and gender are within or outside the models in the primary and secondary groups. SPSS also allows multiple imputation, and we perform it using the same criteria as in MPLUS, with all variables and with 7 datasets.

It is also not clear how you used the effect size measure in lines 315-317.

RE: we remove the effect size for MANOVA since it is not relevant in this case because just a significant effect is enough to consider one variable in the subsequent SEM model. 

 Nor is it clear how you used the missing at random test in lines 318-322. 

RE: Little's MCAR test was used to examined missing values. It was significant in both samples. Anyway, Enders(2010) consider that is not enough exact test.

A clearer explanation of how much data were missing would be helpful. With SEM, maximum likelihood imputation is usually used to handle missing data.

Re: we added this: the maximum number of unanswered items for each participant was less than 20% in both samples. 28 cases in primary and 36 cases in secondary had all the items of perceived discrimination as missing values. 

Multiple imputations and full information maximum likelihood (FIML) are asymptotically equivalent. We chose to use MI because it is better with non-normal distribution, according to Nacagawa (2015) and Enders(2010). All variables were used in each sample to perform multiple imputation of missing values

Please describe the difference between Public and Subsidized schools. 

RE: Public schools are fully funded by the government. While government-subsidized schools only receive partial funding to operate, they are free to charge school fees per pupil. Private schools do not receive any funding; they charge school fees per pupil.

Please provide a reference for the omega statistic. 

RE: We added Hayes & Coutts (2020) reference in the data analysis section

Only scales that use the agree-disagree format are truly Likert scales. Other types of responses are merely rating scales (e.g., never to every day). 

RE: we modified all.

It is not clear why you say that the self-esteem scale is a self-report instrument (line 217) when all of your assessments involve self-report. 

RE: All instrument we used were self-report, but we prefer to specify in each one to be clear

It is not clear what you mean in lines 228-229 about "adjusted in relation to nationality."

RE: collective self-esteem was considerer in national level. We added one example of ítem. The initial instruction of this instruments was "Depending on which nationality you consider yourself, mark your grade according to the following statements."

You should also remove all statements of causal direction in your description of the findings. Rather than saying a variable had a direct effect, you can say that it was directly related. You are also using regression in your models, so path weights are regression weights and not correlations. 

RE: we modified all.

Also, please describe how you tested indirect relations. This is usually done with bootstrapping.

RE: We use the model indirect effect syntaxis in MPLUS. Boostraping is not available for multiple imputation.

Finally, you still have issues with appropriate use of English. I highlighted some places where the English is not clear.

RE: we review and rewrite all these

Best Regards

---

## [Editor Report · Decision Letter 2]

12 Oct 2020

PONE-D-20-05463R2

Perceived discrimination and contextual problems among children and adolescents in Northern Chile

PLOS ONE

Dear Dr. Flores,

Thank you for resubmitting your manuscript to PLOS ONE.  Although you have made improvements in the readability of the paper and have clarified some of the concerns that were raised in the last review, the paper still has serious limitations. I think the results do show that discrimination directed toward students is related to various problems in your schools, but it is not clear that this is a problem with individuals as opposed to differences between schools. All of your findings could result from what are known as between-group differences, which would have very different implications from differences due to individual experiences across schools. Unless you can clarify this in your next revision, should you choose to submit one, it will not be possible to publish this in PLOS ONE. You may want to consult with a data analyst who is familiar with this distinction and who can conduct the necessary analyses.

In addition, you still have a contradiction in your correlation matrices between primary and secondary students in regard to the relation between collective esteem and discrimination. You said this was a mistake, but it's still there in the table.

I still don't understand why age and gender are not in the correlation matrices but are in the SEM. There is no reason not to have them in the correlations.

I must ask you again to not label ethnic identification as ethnic identify. This will confuse most readers, as ethnic identity is merely how one labels on ethnicity. Your measure is more of an esteem measure, which is why it is related to your measure of collective esteem and why it behaves similarly.

I again must ask you to refrain from using causal language in describing your results. Also, I asked you to label the relations in the model as regression coefficients rather than correlations. But you still refer to them as r values.

Finally, I see no reason to present two sets of analyses for the collective problems and the individual components of that measure. It is sufficient to present the results broken out by the subscales, especially since they are so highly related to the overall measure and it is the difference in these that distinguishes your two age groups.

In view of these continued concerns, we invite you to submit a revised version of the manuscript that addresses the points raised above. I again attach a highlighted version of your manuscript where your language needs attention.

Please submit your revised manuscript in the next 30 days. If you will need more time than this to complete your revisions, please reply to this message or contact the journal office at plosone@plos.org. Please include the following items when submitting your revised manuscript:

We look forward to receiving your revised manuscript.

Kind regards,

Daniel Romer

Academic Editor

PLOS ONE

---

## [Author Response · Author response to Decision Letter 2]

8 Dec 2020

Dear editor and reviewers:

Thanks for all your comments. We present a new version of manuscript.

Although you have made improvements in the readability of the paper and have clarified some of the concerns that were raised in the last review, the paper still has serious limitations. I think the results do show that discrimination directed toward students is related to various problems in your schools, but it is not clear that this is a problem with individuals as opposed to differences between schools. All of your findings could result from what are known as between-group differences, which would have very different implications from differences due to individual experiences across

schools. Unless you can clarify this in your next revision, should you choose to submit one, it will not be possible to publish this in PLOS ONE. You may want to consult with a data analyst who is familiar with this distinction and who can conduct the necessary analyses.

RE: We performed a multilevel analysis of two levels. it was necessary to do a path analysis instead of a SEM. The reason was that it is not recommended that there are more parameters than clusters. The results are very similar to those found with SEM.

In addition, you still have a contradiction in your correlation matrices between primary and secondary students in regard to the relation between collective esteem and discrimination. You said this was a mistake, but it's still there in the table.

Re: We remade the entire table, adding age and sex as requested.

I still don't understand why age and gender are not in the orrelation matrices but are in the SEM. There is no reason not to have them in the correlations.

Re: We add both.

I must ask you again to not label ethnic identification as ethnic identify. This will confuse most readers, as ethnic identity is merely how one labels on ethnicity.

Your measure is more of an esteem measure, which is why it is related to your measure of collective esteem and why it behaves similarly.

Re: we change ethnic identity by ethnic identification

I again must ask you to refrain from using causal language in describing your results. Also, I asked you to label the relations in the model as regression coefficients rather than correlations. But you still refer to them as r values.

Re: we change the causal language of the results and conclusions. Also we use the language according to regression coefficients

Finally, I see no reason to present two sets of analyses for the collective problems and the individual components of that measure. It is sufficient to present the results broken out by the subscales, especially since they are so highly related to the overall measure and it is the difference in these that distinguishes your two age groups.

Re: We only present specific models of contextual problems. We eliminate models of global contextual problems.

In view of these continued concerns, we invite you to submit a revised version of the manuscript that addresses the points raised above. I again attach a highlighted version of your manuscript where your language needs attention.

Re: We made changes to the language in the sections indicated

We Will looking forward your answer.

Best regards.

Phd. Jerome Flores Jara

Centro de Justicia Educacional

Universidad de Tarapacá

---

## [Editor Report · Decision Letter 3]

14 Dec 2020

PONE-D-20-05463R3

Perceived discrimination and contextual problems among children and adolescents in Northern Chile

PLOS ONE

Dear Dr. Flores,

Thank you for responding to my concerns in your last submission. I only have a few remaining concerns regarding the way you describe some of your methods, procedures, and findings.

In the Abstract, I’ve highlighted some places where things could be clearer (see attached). “The sample was divided into primary and secondary education groups” would be clearer. “For data analysis, we tested a path analytic model at both the within and between levels to account for the relations between variables.” Not sure what you mean by “and all were optimally adjusted.”

Line 291: you continue to use “ethnic identity”

Line 299: please fix the heading

Line 301: please fix the wording

Lines 352-353:  this is confusing. why report the missing at random test if you are going to ignore it?

Line 356:  not sure what you are saying about "concentrate missing values"

Lines 358: your description of missing data is confusing.  How much missing data did you have? Was it only 28 and 36 cases or was it 45.3% and 36.2%? If the latter, that’s a lot. And if so, I would report what happens if you drop the missing cases and only analyze using list-wise deletion.

Line 359: do you mean the scale was the last one in the survey?

Table 2: please label the collective self-esteem variable the same way in both the figures and the table.

Line 445:  is that p value correct?

Line 544: “visibly” does not make sense.

Line 563: please fix the wording

Line 656: not clear what you are referring to by “second level”.  Please refer to this as the between group level, if that’s what you mean.

We look forward to receiving your revised manuscript.

Kind regards,

Daniel Romer

Academic Editor

PLOS ONE

---

## [Author Response · Author response to Decision Letter 3]

21 Jan 2021

Dear Editor and Reviewers:

Thank you for your valuable comments. We have reviewed the manuscript and made further modifications that we hope will be sufficient.

We have changed the initial theoretical model, since by mistake we had the original general model. Now the specific theoretical model remains.

We check with a native American speaker all modifications in the paper.

Thank you for responding to my concerns in your last submission. I only have a few remaining concerns regarding the way you describe some of your methods, procedures, and findings.

In the Abstract, I’ve highlighted some places where things could be clearer (see attached). “The sample was divided into primary and secondary education groups” would be clearer. “For data analysis, we tested a path analytic model at both the within and between levels to account for the relations between variables.”

Re: we changed each one

 Not sure what you mean by “and all were optimally adjusted.”

Re: we changed for optimum goodness fit

Line 291: you continue to use “ethnic identity”

Re: we changed We had kept it because this is the original explanation of the scale

Line 299: please fix the heading

Re: we changed 

Line 301: please fix the wording

Re: we changed 

Lines 352-353: this is confusing. why report the missing at random test if you are going to ignore it?

Re: we eliminated it to avoid confusion. It was reported as a requirement of the first revisions

Line 356: not sure what you are saying about "concentrate missing values"

Re: it was eliminated, now we only refer to the missing values over the total score

 Lines 358: your description of missing data is confusing. How much missing data did you have? Was it only 28 and 36 cases or was it 45.3% and 36.2%? If the latter, that’s a lot. And if so, I would report what happens if you drop the missing cases and only analyze using list-wise deletion.

Re: we added a paragraph explaining what happens if we only consider the full cases.

Line 359: do you mean the scale was the last one in the survey?

Re: Yes, now we changed

Table 2: please label the collective self-esteem variable the same way in both the figures and the table.

Re: we changed all tables to match with figures

Line 445: is that p value correct?

Re: we changed. It was a mistake

Line 544: “visibly” does not make sense.

Re: we changed the wording

Line 563: please fix the wording

Re: we changed the wording

Line 656: not clear what you are referring to by “second level”. Please refer to this as the between group level, if that’s what you mean.

Re: we changed the wording

Again thank you for your comments.

We look forward to hearing from you

Dr. Jerome Flores Jara

Centro de Justicia Educacional

Universidad de Tarapacá

---

## [Editor Report · Decision Letter 4]

1 Feb 2021

Perceived discrimination and contextual problems among children and adolescents in Northern Chile

PONE-D-20-05463R4

Dear Dr. Flores,

We’re pleased to inform you that your manuscript has been judged scientifically suitable for publication and will be formally accepted for publication once it meets all outstanding technical requirements.

Kind regards,

Daniel Romer

Academic Editor

PLOS ONE

---

## [Editor Report · Acceptance letter]

8 Feb 2021

PONE-D-20-05463R4 

Perceived discrimination and contextual problems among children and adolescents in Northern Chile 

Dear Dr. Flores:

I'm pleased to inform you that your manuscript has been deemed suitable for publication in PLOS ONE. Congratulations! Your manuscript is now with our production department. 

Kind regards, 

on behalf of

Dr. Daniel Romer 

Academic Editor

PLOS ONE